# The association between competitor level and the physical preparation practices of amateur boxers

**Mitchell James Finlay●\*, Richard Michael Page, Matt Greig, Craig Alan Bridge**

Department of Sport & Physical Activity, Sports Injury Research Group, Edge Hill University, Ormskirk, Lancashire, United Kingdom

\* Finlaym@edgehill.ac.uk

## Abstract

Physical training, testing, and monitoring are three key constitutes of athlete physical performance; however, there is a currently a lack of information on the prevalence of such methods in amateur boxing. This study aimed to explore the physical preparation practices of senior elite (SEB) and senior development (SDB) amateur boxers, and to determine whether these practices were discriminated by competitor level. One hundred and one amateur boxers (SEB n = 59, SDB n = 42) were surveyed on their understanding, perceptions and application of physical training, monitoring, and testing practices. SEB were associated with strength/power training (SEB 78%, SDB 50%, P = 0.005), monitor of training intensities (SEB 68%, SDB 40%, P = 0.006), and performing regular fitness testing (SEB 76%, SDB 50%, P = 0.006), compared to SDB. Likewise, SEB were twice as likely (56%) to have their physical preparation managed by a strength and conditioning (S&C) coach or sport scientist, compared to SDB (26%; P = 0.005). For the first time, these data demonstrate the extent to which competitor level is associated with preparatory practices in amateur boxing. Cost was identified as the main barrier in implementing several forms of scientific support in SDB. These data serve as a framework to enhance preparatory practices across different competitor levels in amateur boxing. This might include boxer and coach education on the benefits to a more scientific approach, and the use of cost-effective methods to develop, monitor and assess amateur boxers physical performance. This may be of particular importance where boxers are not funded, such as the SDB in the current study. However, this work may also be used to emphasise the importance of strength/power training, physical fitness testing and monitoring at the elite level of amateur boxing.

## Introduction

Boxing is a combat sport, consisting of intermittent bouts of multi-directional, high-intensity activity, interspersed by 1-min periods of active and passive recovery over a specific number of rounds [1]. Amateur bout activity includes repeatedly striking an opponent ($\geq$ 20 punches per minute) [1,2], evading of punches, and manoeuvring around a boxing ring, with an activity

---

**Data Availability Statement:** All relevant data are within the paper.

**Funding:** The author(s) received no specific funding for this work.

**Competing interests:** The authors have declared that no competing interests exist.

rate for male elite amateur boxers across 3 rounds of ~ 1.31 ($s^{-1}$) [1,3]. This activity can induce a significant biomechanical and physiological demand, in addition to the substantial psychological and hormonal demand associated with combat [3–7]. High levels of aerobic and anaerobic fitness are required in order to sustain repeated high intensity efforts and recover between rounds [8]. Muscular strength, speed, and power are required to produce forceful and high velocity punches [9–11], whilst muscular endurance enables sustained activity [5,8].

The optimisation of the aforementioned physical characteristics is therefore important to a boxers success within the sport [5,8], suggesting the need for regular physical training, monitoring and testing [12]. This could assist in benchmarking fitness data, establishing short- and long-term training goals, and providing motivation to athletes. However, specific research into the physical preparation practices of amateur boxers does not yet exist. Consideration of the level-specific practices, perceptions, understanding and application of scientific methods to physical preparation strategies utilised in amateur boxing, could be valuable in potentially highlighting the need for future applied and research-based interventions. It may be expected that the physical preparation practices of senior elite (SEB) and senior development (SDB) amateur boxers are specific to their competition level, whereby SEB compete at a higher level than their SDB counterparts. Specifically, this could be as a result of potential differences in access to specialist coaches and equipment, bout schedules or the activity profiles associated with competition at different levels [1–3]. Despite this, there have been few concerted attempts to discern the physical preparation practices of different competitor levels in amateur boxing.

Survey based research can effectively gather large amounts of data [13] and have been used previously to investigate the perceptions and practices of strength and conditioning (S&C) coaches and athletes [14–17]. Moreover, surveys can successfully discriminate practices between groups or categorical variables [16]. However, to the authors knowledge only a single study has utilised this method in combat sports [18]. Anecdotal evidence would suggest many amateur boxers do not have access to S&C coaches or sport scientists, therefore a survey targeted at the athletes themselves, as has been done in other individual sport [16,17] would be a more logical choice to obtain this information.

Consequently, the aims of this study were to use survey-based research to investigate the understanding, perceptions, and applications of physical training, monitoring and testing practices of amateur boxers, and to determine whether this is discriminated by competitor level. In line with the limited research in amateur boxing [9,19], it was hypothesised that SEB would be associated with increased reported application of specific physical training modalities, monitoring, and testing practices.

## Methods

### Subjects

One hundred and one amateur boxers completed the survey, including 59 SEB (Mean ± SD sex 50 males 9 females; age 20 ± 3 yrs; mass 71.5 ± 9.7 kg; training experience 7 ± 2 yrs) and 42 SDB (Mean ± SD sex 35 male 7 female; age 21 ± 4 yrs; mass 69.4 ± 9.9 kg; training experience 6 ± 2 yrs). A total of 101 survey completions translated to a completion rate (Number of participants who answered the first question/number of participants who completed the survey) of 88%. Participants were required to be carded senior amateur boxers within their respective governing body in the United Kingdom and be regularly training ≥ 6hrs.per.week. The SEB in this study were defined as those that had competed at their respective National Elite Amateur Boxing Championships as a minimum, and SDB were those who had not. Ethical approval for this study was provided by the University's Research Ethics Committee (SPA-REC-2019-253) and was conducted in accordance with the Helsinki Declaration [20].

## Procedures

The survey 'Physical Preparation Practices of Amateur Boxers' was developed via Google forms software (Google, US), to examine the understanding, perceptions and application of physical preparation strategies of amateur boxers in the UK. Participants were recruited via social media advertisements which linked to an open weblink (Google, US), and in person during visits to boxing gyms in the North of England. The open weblink enabled boxers to read the information on the study procedures, the potential benefits and risks associated with participation, and provide written informed consent. The survey was administered during the 2019–2020 boxing season. Prior to distribution, the suitability and content validity [21] of the questionnaire was assessed by an advisory group of boxers, boxing coaches (National coaching license), S&C coaches (UKSCA) and sport scientists (PhD and MRes).

**Survey topics.** Information on participant characteristics and boxing experience was initially obtained, and participants were also asked if they had access to an S&C coach or sport scientist. The survey was constructed around three main themes which are key constitutes of physical preparatory practices (physical training, monitoring, and testing). Participants were asked to indicate their typical weekly training schedules inclusive of training volumes (weekly hours), modes (technical, circuit, strength/power, conditioning, recovery or other), and session format (group/team or individual) across different phases of competition (typical training week or bout week). Likewise, participants were also asked to specify their understanding and perceptions of select training modes and physical attributes in relation to boxing performance. The survey collected information on the equipment and methods typically used to monitor the intensity of training sessions (Heart rate [HR], rating of perceived exertion [RPE], accelerometery, blood lactate [BLa], other or none). Participants were asked if they routinely performed physical fitness or sport-specific assessments (Aerobic or anaerobic fitness tests–laboratory [LAB] based or non-laboratory [FIELD] based, 1 or 3 repetition maximum [RM] strength test, jump test, upper-body power, velocity-based training, punch force test, mobility assessment/screening, other or no). Participants also specified their perceptions of the benefit of such physical fitness tests.

Lastly, participants were asked to provide their perceptions of the benefits of scientific support to boxing performance on a 5-point Likert scale, and to state their perceived barriers (cost, time, equipment, knowledge or other) to implementing these practices in their overall physical preparation. The content and format (drop-down, multiple-choice, short answer text, checkboxes, tick box and Likert scale) of each individual question in the survey is provided as supplementary information (S1 Table).

## Statistical analyses

A selection of the categorical questions were of a nominal nature and analysed via chi-square tests for association [22]. In instances where expected cell frequencies were less than 5, a Fischer's exact test was performed as an alternative [22,23]. Questions of an ordinal nature were analysed via a Mann-Whitney U test. As part of the assumptions of the Mann-Whitney U test, the distribution shapes of the two groups of the independent variable were considered, and analysis of median scores was possible. These tests were selected to determine possible associations or differences in the physical preparation methods according to competitor level. All statistical analyses were conducted with SPSS 25.0 for Mac (SPSS Inc, Chicago, USA) with significance level was set at $p < 0.05$. Phi (φ) was consulted to determine the strength of associations in the instance of a statistically significant difference found in the chi-square tests [23]. Where applicable, descriptive statistics were also provided for descriptive comparison between groups.

## Results

There was a statistically significant association between competitor level and access to an S&C coach or sport scientist ($\chi^2(1) = 7.894$, P = 0.005), with descriptive data demonstrating that SEB (54%) had greater access to these practitioners when compared to their SDB (26%) counterparts. Less than half of all boxers (42%) had regular access to an S&C coach or sport scientist.

### Physical training

Median training hour scores during bout week were significantly higher in SEB (8hrs.per. week) when compared to SDB (7hrs.per.week), (U = 1557.5, z = 2.243, P = 0.025). No significant differences were identified between groups in the median training hour scores in a typical training week (U = 1.478.4, z = 1.672, P = 0.095). Total training hours for all boxers were significantly (P < 0.001) lower during bout week (Median = 8) when compared to a typical training week (Median = 10), indicative of a tapering of workload.

Table 1 comprises data on the association between competitor level and the training modes and structure during a typical training week, and during the week of a bout. Descriptive data was also included for comparisons between groups.

There was a significant association between competitor level and the perception that circuit training adequately improved strength ($\chi^2(1) = 5.385$, P = 0.020, $\varphi = .231$), with descriptive data indicating SDB were more likely to perceive that circuit training adequately induces strength adaptations (Fig 1). More than half of all boxers (56%) stated they believed circuit training improved strength. There was no significant association between competitor level and all other physical attributes (anaerobic fitness $\chi2(1) = .279$, p = .597; aerobic fitness P = 0.519; muscular endurance $\chi^2(1) = .037$, P = 0.848; speed $\chi^2(1) = 1.931$, p = .165; agility $\chi^2(1) = 1.130$, P = 0.288; power $\chi^2(1) = .209$, P = 0.648, other $\chi^2(1) = .059$, P = 0.807).

Table 2 comprises data on the association of the perceived importance of physical attributes, and competitor level. Descriptive data was also included for comparisons between groups. Half of all boxers (50%) selected aerobic fitness as the most important physical quality to boxing performance, the most popular choice in both groups.

**Table 1. Training structure and modes during a typical week and bout week.**

| Training structure | SEB n (%) | SDB n (%) | $\chi^2$ | P-value | $\varphi$ |
|---|---|---|---|---|---|
| Individual | 23 (39%) | 8 (19%) | 4.584 | .032* | .213 |
| Group/team | 36 (61%) | 34 (81%) | | | |
| Typical week training modes | | | | | |
| Technical | 59 (100%) | 42 (100%) | - | - | - |
| Conditioning | 59 (100%) | 39 (93%) | 4.343 | .069 | - |
| Circuits | 55 (93%) | 42 (100%) | - | .139 | - |
| Strength/power | 46 (78%) | 21 (50%) | 8.583 | .005* | - |
| Other | 3 (5%) | 3 (7%) | - | .687 | - |
| No training days | 59 (100%) | 41 (98%) | .718 | .397 | - |
| Bout week training modes | | | | | |
| Technical | 59 (100%) | 42 (100%) | - | - | - |
| Conditioning | 56 (95%) | 35 (83%) | - | .088 | - |
| Circuits | 34 (58%) | 30 (71%) | 2.013 | .156 | - |
| Strength/power | 26 (44%) | 13 (31%) | 1.780 | .182 | - |
| Other | 3 (5%) | 1 (2%) | - | .446 | - |
| No training days | 59 (100%) | 41 (98%) | - | .416 | - |

*denotes significant association with competitor level (P $\leq$ 0.005).

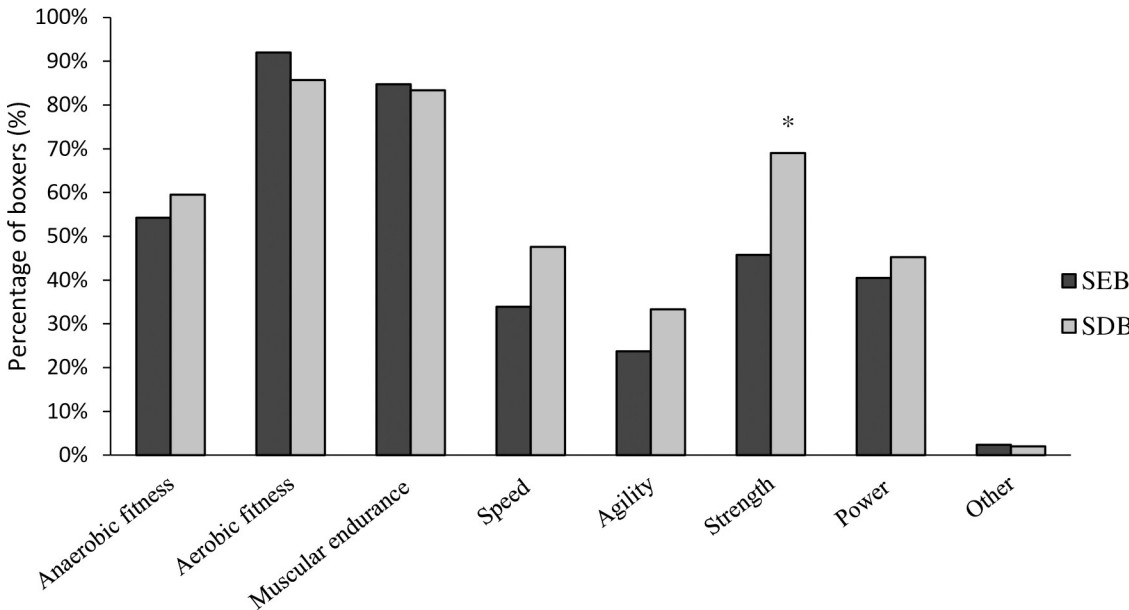

**Fig 1. Percentage of boxers who perceive circuit training improves specific physical fitness attributes.** *denotes significant association between competitor level and perception of physical fitness to circuit training (P ≤ 0.005).

### Monitoring

There was a significant association between competitor level and the monitoring of training intensity ($\chi^2(1)$ = 7.449, P = 0.006, φ = .272), with descriptive data indicating a reported lower application in SDB (40%) when compared to SEB (68%) (Fig 2). The most frequently used method for all boxers was HR monitoring (42%). There were significant associations between competitor level and the use of both HR monitoring ($\chi^2(1)$ = 12.024, P = 0.001, φ = .345) and RPE ($\chi^2(1)$ = 4.117, P = 0.042, φ = .202), with descriptive data indicating a greater use in SEB (56%; 29%), when compared to their SDB counterparts (21%; 12%). There were no statistically significant associations between competitor level and the use of all other methods (boxing-specific accelerometery $\chi^2(1)$ = 2.104, P = 0.147; BLa $\chi^2(1)$ = 2.965, P = 0.085).

### Testing

There was no significant associations between competitor level and the perception that regular fitness testing is beneficial to boxing performance ($\chi^2(1)$ = .794, P = 0.373), with descriptive data indicating similar positive agreement in both groups (SEB 93%, SDB 88%). As identified

**Table 2. Number of boxers who perceived physical attributes as the most important to boxing performance.**

| Physical Attribute | SEB (n) | (%) | SDB (n) | (%) | χ2 | P-value | φ |
|---|---|---|---|---|---|---|---|
| Aerobic fitness | 24 | 42% | 27 | 65% | 5.470 | .019* | .233 |
| Anaerobic fitness | 12 | 20% | 6 | 14% | .614 | .433 | - |
| Speed | 9 | 15% | 5 | 11% | .535 | .464 | - |
| Muscular endurance | 5 | 8% | 2 | 4% | .524 | .469 | - |
| Power | 5 | 8% | 1 | 3% | - | .395 | - |
| Strength | 4 | 7% | 1 | 3% | - | .399 | - |

*denotes significant association between competitor level and perceived importance of physical attributes (P ≤ 0.005).

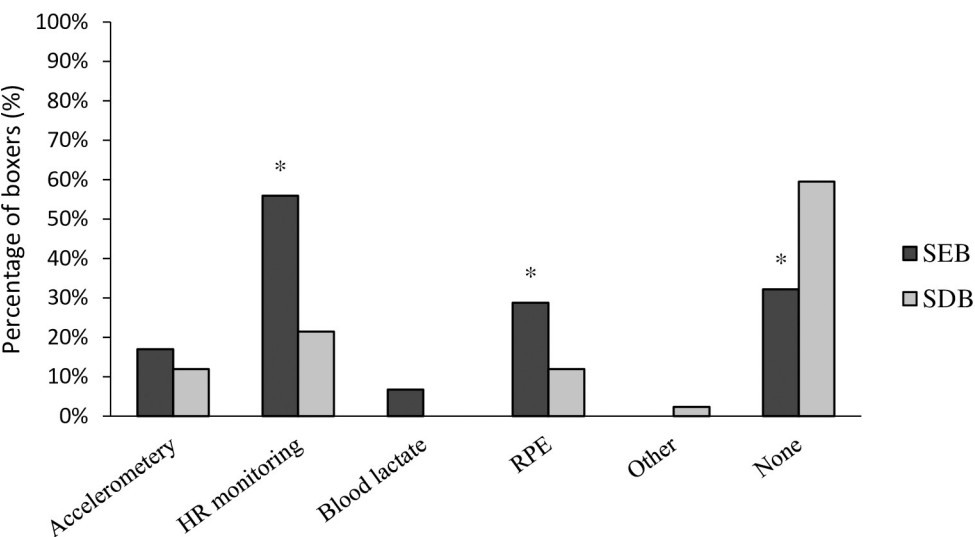

**Fig 2. Percentage of boxers who monitored training intensity.** *denotes significant association between competitor level and method of monitoring training intensity (P ≤ 0.005).

in Fig 3, there was a significant association between competitor level and the regular use of physical testing ($\chi^2(1)$ = 7.478, P = 0.006, φ = .272) with descriptive data showing a lower reported application in SDB (50%) when compared to SEB (76%). More than half (65%) of all boxers regularly took part in physical fitness testing. Field-based aerobic fitness tests were the most frequently reported test used (50%) by all boxers. There were statistically significant associations between competitor level and the completion of several physical fitness tests (aerobic fitness tests [LAB] $\chi^2(1)$ = 9.293, P = 0.002, φ = .303; 3/5 rep max strength tests $\chi^2(1)$ = 5.852, P = 0.016, φ = .241; upper body power tests $\chi^2(1)$ = 5.543, P = 0.019. φ = .234; jump tests $\chi^2(1)$

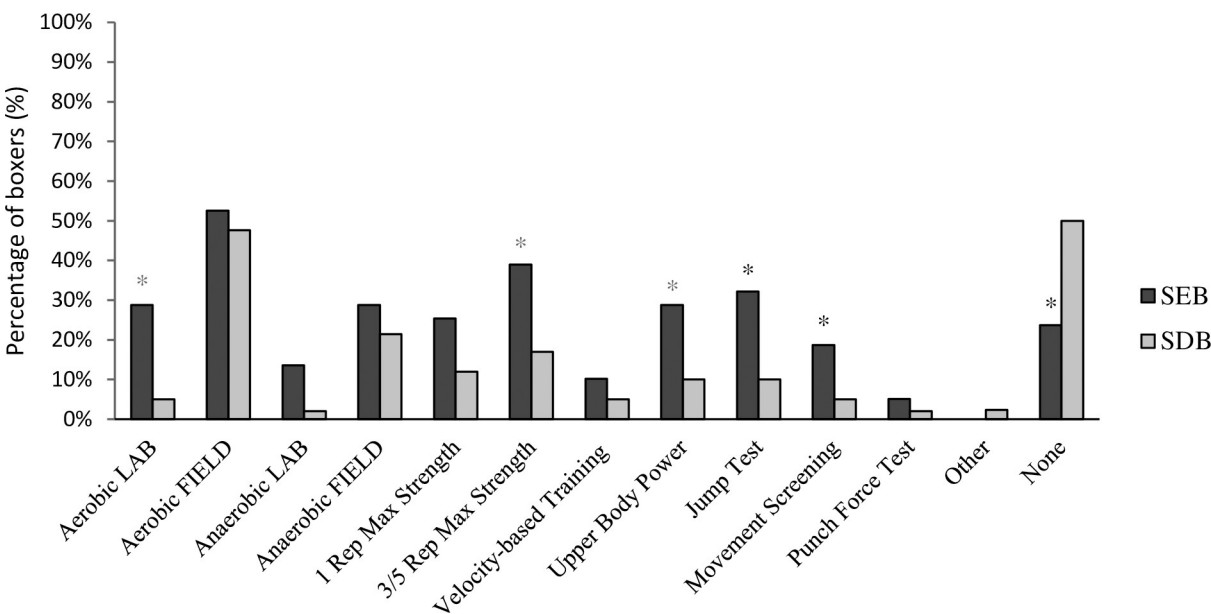

**Fig 3. Percentage of boxers who regularly performed common physical fitness tests.** *denotes a significant association between competitor level and the use of physical fitness tests (P ≤ 0.005).

= 7.176, P = 0.007, φ = .267; movement screening $\chi^2(1)$ = 4.216, P = 0.040, φ = .204). There were no statistically significant associations between competitor level and the completion of all other physical fitness tests (anaerobic fitness tests [FIELD] $\chi^2(1)$ = .700, P = 0.403; aerobic fitness tests [FIELD] $\chi^2(1)$ = .238, P = 0.626; anaerobic fitness tests [LAB] P = .076; 1 rep max strength tests $\chi^2(1)$ = 2.824, P = 0.093; velocity-based training (VBT) P = 0.321; field-based aerobic fitness tests P = 0.321; punch force test P = 0.492; other P = 0.234). Descriptive data demonstrated a consistently higher reported application in physical fitness testing in SEB when compared to their SDB counterparts (Fig 3).

## Perceptions of scientific support and potential barriers

There were no statistically significant differences in median Likert scale scores between competitor level and the perceived benefit of scientific support to boxing performance (U = 1483.5, z = -1.895, P = .058). Descriptive data suggest SEB were more likely to 'strongly agree' that scientific support benefits boxing performance (68%), when compared to SDB (48%) (Fig 4).

There was a statistically significant association between competitor level and cost being a barrier to adopting a scientific approach to physical preparation (SEB 46%, SDB 74%, $\chi^2(1)$ = 7.894, P = 0.005, φ = .280). More than half of all boxers (57%) identified cost as a barrier to the implementation of scientific support. There were no statistically significant associations between competitor level and all other perceived barriers (time: SEB 37%, SDB 48%, $\chi^2(1)$ = .034, P = 0.854; equipment: SEB 46%, SDB 50%, $\chi^2(1)$ = .177, P = 0.674; knowledge: SEB 44%, SDB 55%, $\chi^2(1)$ = 1.123, P = 0.289).

## Discussion

This study aimed to explore the physical preparation practices of amateur boxers, and to determine whether these practices are discriminated by competitor level. In agreement with the studies hypothesis, competitor level was significantly associated with the inclusion of strength/power training, and utilising monitoring and physical fitness testing practices. Descriptive data showed that SEB boxers were more likely to include strength/power training in their physical training and utilise monitoring and fitness testing practices compared to SDB. Further, SEB were twice as likely to have their physical preparation managed by performance staff

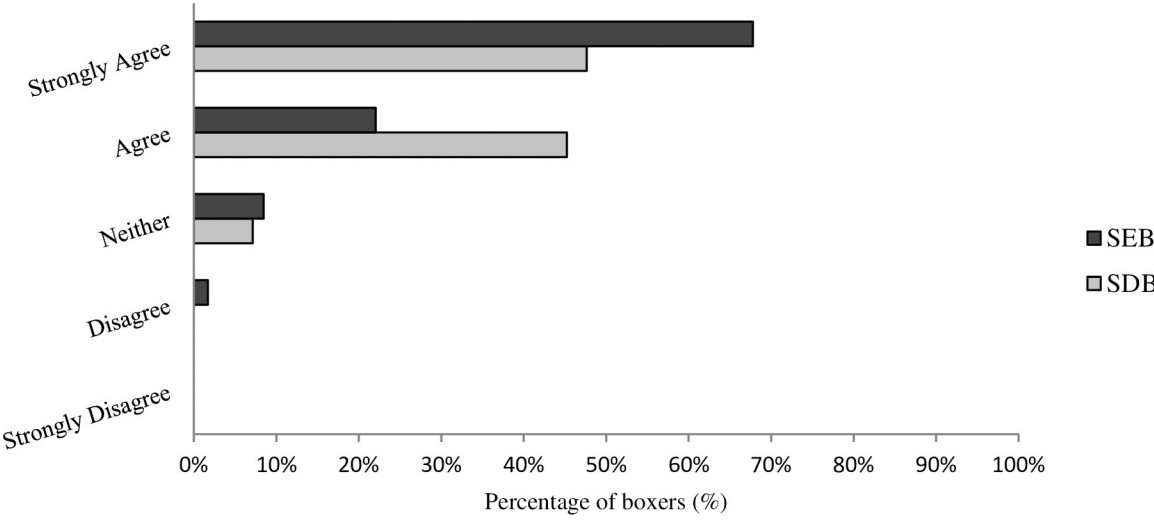

**Fig 4. Likert scale scores on whether scientific support benefits boxing performance.**

such as an S&C coach or sport scientist, compared to SDB. This might be expected as the SEB group including many funded athletes who are part of national squads, or scholar athletes.

No differences were observed in typical weekly training hours for both groups of boxers (~10hrs.per.week), however SEB performed a greater volume in bout week (8hrs.per.week), compared to SDB (7hrs.per.week). Training hours decreased during bout week by 2 and 3 hours for SEB and SDB respectively, indicative of a tapering of workload in both groups. The values of weekly training hours in this study are within the range previously reported in a case study on a professional boxer (7-12hrs.per.week) [24], and the most frequently reported in Italian amateurs (6-10hrs.per.week) [25]. Any tapering of workload in combat sports must be balanced against weight-making practices commonly executed by athletes, whereby a similar or even an increased amount of activity may be performed in the days leading up to a bout [26]. The majority of SEB and SDB retained conditioning sessions during bout week, suggesting it is possible boxers in this study could have been partaking in this well acknowledged practice. The activity performed during a typical week did not differ between the groups, with the exception of strength/power training, which was more prevalent in SEB, compared to SDB. Resistance training has traditionally been associated with negative detriments to speed and movement in boxing performance [9], however it is known that when programmed correctly, weightlifting and Olympic lifting derivatives can enhance athletic performance [10,27]. No optimal frequency of strength/power training for athletes currently exists in the literature, however twice weekly sessions performed by participating boxers in this study, have been shown to be effective at developing maximal strength in athletes when intensity is at 85% 1RM [28]. Considering the benefits to performance and injury reduction associated with strength training [27,29], it is surprising that 22% of SEB boxers and half of SDB boxers did not partake in this form of training. Clearly a greater application of resistance training aimed at increasing strength and power qualities, would improve boxing performance and assist in the reduction of training time lost due to injury.

More than 90% of all boxers, irrespective of standard, completed technical, conditioning and circuit-based training modes. Circuit training has long been present in a boxers training regime, when asked which physical attributes boxers primarily improve as a result of circuit training, a greater percentage of SDB chose strength and power (69%, 45%) compared to SEB (46%, 40%). This could suggest that many boxers believe they are already obtaining sufficient strength and power improvements through more traditional means, to meet the demands of the sport. Boxing-specific circuit training can improve aerobic capacity and induce slight improvements in strength amongst other physical attributes [30]; however, it may not necessarily develop the rapid force production abilities and proximal to distal sequencing essential for forceful movements such as punching [11,31–34]. Considering the perceptions of many amateur boxers, additional support and education on the scientific principles and benefits to select training modes, particularly in SDB, who may not have access to performance staff, would improve physical preparation in amateur boxing. This is reflected in the low overall support for strength or power as the most important physical qualities to boxing performance, perhaps justifiably so.

The current study also considered the association between competitor level and the scientific methods used to monitor training intensities. SEB were nearly twice as likely to use scientific methods to monitor training intensity, in comparison to SDB. Heart rate (HR) monitoring was the most frequently used method to monitor training intensity, with 56% of SEB and 21% of SDB utilising this method respectively. HR monitoring is an inexpensive, time-efficient and non-invasive method [35] to assess cardiovascular demands and monitor exercise intensity in combat training, simulated bouts and in competition [3,4,36,37]. Rating of perceived exertion (RPE) is a subjective, non-invasive, and valid method of quantifying

internal training and combat loads in combat athletes [6]. Similar to HR, a greater percentage of SEB (29%) used RPE when compared to SDB (12%). Previous research has demonstrated a strong relationship between RPE and HR during select combat activity [6]. Additionally, the use of RPE appears to be sensitive enough to detect fatigue induced changes in the perception of exertion during simulated boxing combat [4]. A small percentage of boxers in both groups revealed they used boxing-specific accelerometery or 'punch trackers' to monitor striking quality, classification and performance. In recent years there has been an increased research focus on the use of inertial sensors and linear position transducers in combat sport [38–41]. This technology has been used to quantify whole body external loads [4,36,41] and punch-specific performance [39,40]. To what extent the more commercially available punch trackers can be used effectively to monitor punch-specific performance, is dependent on the validity and reliability of such devices [38]. Linear position transducers such as the GymAware (GymAware, AUS) have been shown to be reliable in measuring punch velocity in boxers [39,40]; however, they typically incur a great cost. Nevertheless, the above equipment could be valuable in quantifying punch output and performance in the absence of video analysis. Amateur boxers could utilise some of the aforementioned inexpensive and non-invasive methods to monitor the internal and external response to training and simulated bouts, if reliability and validity has been established.

This work was also concerned with the association between competitor level and physical fitness testing in amateur boxing. The majority of all boxers perceived regular fitness testing to be beneficial to boxing performance, however, SEB were more likely to participate in fitness testing, when compared to SDB. Moreover, the tests performed varied between competitor level. SEB were generally more likely to perform all conditioning fitness tests in the laboratory when compared to SDB. Specific to conditioning, laboratory-based aerobic and anaerobic tests were more prevalent in SEB (29%, 14%) when compared to SDB (5%, 2%), though a significant association between competitor level was only found in the laboratory-based aerobic tests. In contrast, the more accessible field-based aerobic and anaerobic alternatives were much more prevalent in SDB (48%, 21%), however this was still slightly below SEB levels (53%, 29%). Boxing is predominantly aerobic in nature, albeit with anaerobic contributions [1,5], perhaps indicating the appropriateness of such tests in a boxers training programme. Previous literature shows amateur boxers performance in laboratory-based aerobic tests indicate a good level of aerobic capacity (49 and 65 ml·kg$^{-1}$·min$^{-1}$) [8]. Amateur boxers could utilise sprints and intermittent field-based, accessible tests such as the Yo-Yo IRT to assess anaerobic and aerobic fitness levels, previously included in an amateur boxing testing battery [12]. Fitness tests relating to muscular strength, power and impulse followed a similar trend as the conditioning tests, with SEB in this study being between 2-3x more likely to perform upper and lower body assessments of the above physical qualities. Repetition maximum tests and velocity-based training (VBT) were both more prevalent in SEB boxers, possibly suggesting a greater monitoring of resistance training performance and informed training prescription in elite boxers, when compared to their more novice counterparts. The CMJ and squat jump tests both show good reliability in assessing lower body impulse in youth and senior amateur boxers [12]. Recent developments in mobile applications have enabled the low-cost and simple assessment of jumping performance from a mobile or tablet device [42]. Likewise, upper body tests often included in a boxers training or testing battery, such as the landmine throw, and medicine ball throw have received attention in the literature [5,12,43]. The latter has recently been used as a more accessible, field-based alternative to the direct measurement of punch force against a force plate [12]. Perhaps as expected, only 4% of all boxers in this study performed direct assessments of punch force, with a slightly greater prevalence exhibited in SEB. Whilst vertically-mounted force plates and punch integrators may be viewed as the most ecologically valid

measure of punch force [44,45], the data in this study confirms the rarity of this equipment in combat sports. Movement screening was also more prevalent in SEB when compared to SDB. Whilst a greater application of fitness testing was apparent in SEB, it remains lower than it perhaps should be. Boxers and their coaches could incorporate testing protocols to quantify fitness levels, increase motivation and better-inform training prescription. Particularly, boxers could take advantage of some of the above tests [5,12,42,43] that do not require laboratory access and extensive equipment once reliability and validity has been established.

The benefit of scientific support to a boxers physical preparation was universally supported across both groups, although SEB were more inclined to strongly agree when compared to SDB. Despite this, differences in the perceived barriers to implementing scientific support varied between groups. The cost of scientific support as a barrier was more prevalent in SDB, compared to SEB. This can be linked to the finding that SEB were much more likely to have access to an S&C coach or sport scientist. The less frequent use of strength/power training, monitoring, and testing in SDB, might be chiefly attributable to a lack of access and opportunity, compared to SEB. It is hoped that the recommendations outlined in this study could therefore have important implications for SDB, and indeed SEB who may not receive funding or regular access to performance staff.

## Strengths and limitations

The main strength of the current study is that it provides novel data on physical training, testing, and monitoring practices in amateur boxing. Likewise, to the authors best knowledge, it provides the first data on boxer perceptions of scientific support to boxing performance. However, there are also limitations that should be noted. Perhaps the primary limitation, though this is to be expected, is the potential differences in self-reported information compared to objective measurement of variables [46]. An additional limitation is the small sample size for a survey-based study; however, the authors acknowledge the difficulties of obtaining large sample sizes in an amateur boxing population. Likewise, there was a lack of female participants in the study, compared to males. Future research may wish to aim for a more proportionate sample of males and females, in order to assess potential differences in the physical preparation practices between sexes.

## Conclusion

The current data demonstrate that elite level amateur boxing is associated with a greater reported application of a number of physical preparation practices, including strength/power training, physical fitness testing, and monitoring practices. This is despite no differences in the perceived value of scientific support between SEB and SDB. Less frequent application of strength/power training, monitoring and fitness testing in SDB appear chiefly attributed to two primary barriers which interlink; cost, and access to performance staff. Thus, this study sought to provide potential cost-effective, valid and reliable scientific methods that can be applied to a boxers physical preparation. Amateur boxers may wish to consult a qualified S&C coach or exercise scientist and partake in strength and power training where possible. Likewise, amateur boxers could utilise the inexpensive and non-invasive methods to monitor training intensity and assess physical fitness, thus having implications for informed training design. These findings suggest researchers and practitioners should aim to educate boxers and coaches in the value of scientific approaches to physical preparation, and assist in dispelling long-standing myths, such as negative perceptions of resistance training in boxing. The addition of testing and monitoring practices could also enhance the performance of the amateur boxer, by setting benchmark physical goals and increasing motivation.

## Supporting information

**S1 Table. Physical preparation practices of amateur boxers survey.** Survey questions. (DOCX)

## Acknowledgments

The authors would like to thank the amateur boxers for their time in completing the survey and wish them luck in their boxing careers.

## Author Contributions

**Conceptualization:** Mitchell James Finlay.

**Data curation:** Mitchell James Finlay.

**Formal analysis:** Mitchell James Finlay.

**Investigation:** Mitchell James Finlay.

**Methodology:** Mitchell James Finlay.

**Supervision:** Richard Michael Page, Matt Greig, Craig Alan Bridge.

**Visualization:** Mitchell James Finlay.

**Writing – original draft:** Mitchell James Finlay.

**Writing – review & editing:** Richard Michael Page, Matt Greig, Craig Alan Bridge.

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
