## [Decision Letter · Decision Letter 0]

6 Jul 2021

PONE-D-21-10978

The Influence of Competitor Level on the Physical Preparation Practices of Amateur Boxers.

PLOS ONE

Dear Dr. Finlay,

Thank you for submitting your manuscript to PLOS ONE. After careful consideration, we feel that it has merit but does not fully meet PLOS ONE’s publication criteria as it currently stands. Therefore, we invite you to submit a revised version of the manuscript that addresses the points raised during the review process.

ACADEMIC EDITOR

Dear Authors,

the manuscript has been revised by two experts in the filed that retrieved several major points that you should reply while revising your article.

Please take into consideration all the points in particular to those referring to the methodological approach and results presentation.

We look forward to receiving your revised manuscript.

Kind regards,

Emiliano Cè

Academic Editor

PLOS ONE

Journal Requirements:

Reviewers' comments:

Reviewer's Responses to Questions

**Comments to the Author**

1. Is the manuscript technically sound, and do the data support the conclusions?

Reviewer #1: Yes

Reviewer #2: Partly

2. Has the statistical analysis been performed appropriately and rigorously? 

Reviewer #1: N/A

Reviewer #2: Yes

3. Have the authors made all data underlying the findings in their manuscript fully available?

Reviewer #1: Yes

Reviewer #2: Yes

4. Is the manuscript presented in an intelligible fashion and written in standard English?

Reviewer #1: Yes

Reviewer #2: Yes

5. Review Comments to the Author

Reviewer #1: The Influence of Competitor Level on the Physical Preparation Practices of Amateur Boxers.

GENERAL CONSIDERATIONS

Authors aimed to investigate the possible differences in understanding, perceptions and applications of physical training, monitoring and testing practices between amateur and senior boxer.

The study is well organized and with an interesting aim. However, some corrections are needed.

- Please, delete the space in Lines 62, 77, 86, 93, 95, 104, 114, 118, 122, 129, 133, 137, 142, 162, 168, 176, 184-196, 200, 211, 214, 221, 225, 237, 240, 260, 263, 268, 271, 279-281, 291, 316, 333, 360, 399, 410.

SPECIFIC CONSIDERATIONS

ABSTRACT AND INTRODUCTIONS

- Please, specify the acronym S&C in the abstract (line 34) and in introduction (line 79) and delete it in Physical Training paragraph (line 123)

METHODS

- Please, specify the University of the Research Ethics Committee and the approval number if provided.

- Authors state that SEB were defined as those that had competed at their respective National Elite Amateur Boxing Championships as a minimum, and SDB were those who had not. This classification was based on previous studies, or it is a formal classification based on the rules of an international boxing federation?

Subject paragraph:

- please, provide the reference for the Helsinki Declaration

Survey paragraph:

- please, delete it. The information is just reported in the previous paragraph.

Survey topics:

- For a smoother reading of this and the following paragraphs I suggest writing this information in just one paragraph. It could be less schematic and more discursive.

Physical Training:

- Please, see the comment above on the S&C acronym

Statistical Analyses

- Why authors used Mann-Whitney U test to compare the frequency of the opinions expressed by subject on the benefit of S&C and scientific support to boxing performance?

- Why authors used Friedmans ANOVA test if they have two groups?

Reviewer #2: GENERAL COMMENTS

Dear authors,

Thank you for this manuscript that sheds light on the differences between two competitors levels in boxing.

I appreciated the paper; however, I found some difficulties in interpreting and understanding the numerous variables, probably because the survey explanation is very limited. I suggest improving it.

Furthermore, some data are present in the discussion section without an appropriate exposition in the results section. I suggest the authors report the essential results in the proper section and then comment on them in the discussion one.

Finally, the authors run an ANOVA analysis, but they spoke about influence and association instead of difference. Maybe they should be more precise with correct words or, alternatively, better explain the statistical analysis.

ABSTRACT

I am not sure the authors evaluated the influence, but only the difference between the two groups.

ABSTRACT

The background is missing.

INTRODUCTION

1. Line 52: does this description only refer to amateur boxing?

2. Line 71: could the authors explain better the difference between SEB and SDB. Are these two groups amateur boxers or SEB are considered professional boxers?

3. Line 79: the abbreviation has not been previously introduced.

MATERIALS AND METHODS

1. How have the participants been recruited?

2. Which has been the adherence or the percentage of respondents compared to the potential and contacted participants?

3. Line 111: referring to the recruitments period, could Covid restriction have influenced the results?

4. Survey: in my opinion, the survey should be explained more in-depth; otherwise, it is difficult to understand all the variables explained in the results. For example, in the monitoring section, the authors should give more information about the option present in the survey and the possible responses (frequency or number of hours of the methods’ use, etc). Moreover, in the testing section, the authors should specify what they meant with LAB and FIELD. Finally, for the perception of the scientific support, it is better to specify that it is a Likert scale and the explanations of the points (see later for further details).

5. Statistical analysis: why did the authors decide to utilize the ANOVA analysis? There are only two groups, and the authors even not evaluate neither main effects nor interactions (in the case of a two-way ANOVA).

RESULTS

1. Lines 164-167: from which questions did the authors obtained these data?

2. Are there any other data, such as age, experience years, age at the onset of the sports practice, that could explain or be useful for the data interpretation?

3. The authors differentiated between bout and typical training week without specifying this in the statistical analysis or in the survey explanation (if the questions were double for bout and training week).

4. Lines 178-179: the authors should not repeat data already present in the table. This also happens in other sections of the results.

5. Table 1: the authors could merge the columns of number and percentage in a single column (n (%)).

6. Line 201: in this line and in other sentences, the authors spoke about association; however, Mann-Withney and ANOVA evaluated the differences between the two groups and not the association. On the contrary, they should better specify the run analysis to understand why the wrote association.

7. Line 209-210: this question is not present in the survey explanation.

8. Lines 227-229: these data are not visible in Figure 1.

9. Monitoring section: Data about RPE and none are not explained; however, many comments are present about this topic in the discussion section.

10. Testing: in figure 3, other significances are not commented on in the text.

11. Lines 272-279: this question is not present in the survey explanation.

DISCUSSION

1. Lines 293, 294, 319, 341, and 376-382 report data not present or different from the result section.

2. Line 361: the authors wrote influence, but they evaluated the difference.

3. Lines 366-368: the authors should specify that the data are not significant.

4. Strengths and limitations are missing.

6. PLOS authors have the option to publish the peer review history of their article (what does this mean?). If published, this will include your full peer review and any attached files.

Reviewer #1: **Yes: **Antonino Mulè

Reviewer #2: **Yes: **Lucia Castelli

---

## [Author Response · Author response to Decision Letter 0]

9 Aug 2021

Dear Emiliano Ce,

Thank you to you and your editorial team for the opportunity to submit a revised version of the manuscript. We would also like to send our appreciation to the two reviewers for their time spent on the manuscript and for the quality of their remarks and suggestions. The authors responses to reviewers’ comments are in a point-by-point format, whereby the authors have thoroughly considered each individual comment. This is reflected in the considerable track changes to the manuscript. The authors believe the comments have been particularly helpful in improving the quality of the manuscript, particularly now that the paper is more focused and consistent regarding the terminology used. The authors hope that the considerable work carried out in revising the paper, is of a similarly high standard.

Once again, thank you.

Editors comments

Authors reply – The lead author has ensured the journal style requirements have been adhered to.

Authors reply – The authors have made considerable changes to the survey information. This includes more information in text, but we have also provided a supplementary file which includes the content and the format of each question included in the survey. In agreement with reviewer 1 feedback, the authors have condensed the survey topic section, and have instead included details on the question content and format of each question as a supplementary file (S1 Appendix). It is hoped that the above changes will assist in the interpretation and replication of the study.

Reviewer 1 comments

GENERAL CONSIDERATIONS

Authors aimed to investigate the possible differences in understanding, perceptions and applications of physical training, monitoring and testing practices between amateur and senior boxer. The study is well organized and with an interesting aim. However, some corrections are needed.

- Please, delete the space in Lines 62, 77, 86, 93, 95, 104, 114, 118, 122, 129, 133, 137, 142, 162, 168, 176, 184-196, 200, 211, 214, 221, 225, 237, 240, 260, 263, 268, 271, 279-281, 291, 316, 333, 360, 399, 410.

Authors reply – The authors wish to express gratitude to the reviewer for their kind words about the study. The authors apologise for line number formatting error, the above changes have been made.

SPECIFIC CONSIDERATIONS

ABSTRACT AND INTRODUCTIONS

1. - Please, specify the acronym S&C in the abstract (line 34) and in introduction (line 79) and delete it in Physical Training paragraph (line 123)

Authors reply – The authors have introduced the acronym ‘S&C’ in the abstract and introduction. The acronym in the Physical Training paragraph has been deleted. 

METHODS

2. - Please, specify the University of the Research Ethics Committee and the approval number if provided.

Authors reply- The relevant ethics reference number has now been provided.

3. - Authors state that SEB were defined as those that had competed at their respective National Elite Amateur Boxing Championships as a minimum, and SDB were those who had not. This classification was based on previous studies, or it is a formal classification based on the rules of an international boxing federation?

Author reply – The authors thank the reviewer for their question. The classification is based on the regulations of England Boxing (2019), the leading organisational body of the four-nations (United Kingdom). The classification of SEB and SDB are very similar between England, Wales, Scotland and Northern Ireland. The authors have provided clarification for the boxer classification. Lines 71 – 76 now has additional information on the differences between the two classifications.

Subject paragraph:

4. - please, provide the reference for the Helsinki Declaration

Authors reply – The World Medical Association reference has been provided.

Survey paragraph:

5. - please, delete it. The information is just reported in the previous paragraph.

Authors reply – The authors acknowledge that this section may not have been needed. This has been deleted.

Survey topics:

6. - For a smoother reading of this and the following paragraphs I suggest writing this information in just one paragraph. It could be less schematic and more discursive.

Authors reply - The authors appreciate the reviewer’s suggestion, and we have opted to condense the information into one small paragraph. The authors have also provided supplementary information (S1 Appendix) on the content and format of each individual question, for further clarity.

Physical Training:

7. - Please, see the comment above on the S&C acronym

Authors reply – The above change has been made.

Statistical Analyses

8. - Why authors used Mann-Whitney U test to compare the frequency of the opinions expressed by subject on the benefit of S&C and scientific support to boxing performance?

Authors reply – The authors initially wanted to test for differences in the scores using an independent t-test; however, when running the tests, the normal distribution assumption was violated. The authors decided to use the Mann-Whitney U Test as an appropriate non-parametric alternative to test for differences in Likert scale (Ordinal) data (O’Donoghue, 20212; Armour & Macdonald, 2012, p327). In this instance, the dependent variable is the ‘perceptual benefit of S&C and scientific support’, and the independent variable (2 groups) was ‘boxer level’ (SEB or SDB). The authors have included the relevant reference in the text.

9. - Why authors used Friedmans ANOVA test if they have two groups?

Authors reply- The authors thank reviewer 1 for highlighting this error. The Friedmans ANOVA was initially trialled as a suitable statistical test for the ‘training hours’ question, but this was changed. Much of the text relating to the ANOVA was deleted; however, one sentence was mistakenly left in the manuscript. This has now been deleted.

Reviewer 2 comments

GENERAL COMMENTS

Thank you for this manuscript that sheds light on the differences between two competitors levels in boxing.

I appreciated the paper; however, I found some difficulties in interpreting and understanding the numerous variables, probably because the survey explanation is very limited. I suggest improving it.

Furthermore, some data are present in the discussion section without an appropriate exposition in the results section. I suggest the authors report the essential results in the proper section and then comment on them in the discussion one.

Finally, the authors run an ANOVA analysis, but they spoke about influence and association instead of difference. Maybe they should be more precise with correct words or, alternatively, better explain the statistical analysis.

ABSTRACT

1. I am not sure the authors evaluated the influence, but only the difference between the two groups.

Authors reply - The authors explored the association between competitor level and many physical preparation practices. The authors also provided descriptive statistics (Mean ± SD, %) for comparisons, and tested for differences in select questions. The title and the text throughout the manuscript has been changed to reflect the primary investigation into the association between competitor level and physical preparation practices. The word ‘influence’ may not have been the best choice and has been deleted. 

ABSTRACT

2. The background is missing.

Authors reply – The authors have included the following as an opening background sentence in the abstract:

“Physical training, testing and monitoring are three key constitutes of athlete physical performance; however, there is a currently a lack of information on the prevalence of such methods in amateur boxing.”

INTRODUCTION

1. Line 52: does this description only refer to amateur boxing?

Authors reply – The description would also relate to professional boxing. The authors have changed the start of this sentence. The authors have also added “Amateur bout activity” in the following sentence to introduce the reader to the specific study topic, ‘amateur boxing’.

2. Line 71: could the authors explain better the difference between SEB and SDB. Are these two groups amateur boxers or SEB are considered professional boxers?

Authors reply – The authors have changed the structure of this paragraph. Specifically, the authors have added:

“whereby, SEB compete at a higher level than their SDB counterparts.” 

We feel that this now introduces the differences between the two competitor levels, which is presented in greater detail in the methods section. We have also included the words amateur boxing, to directly answer the reviewers query on whether SEB related to professional boxing. We hope these changes are adequate. 

3. Line 79: the abbreviation has not been previously introduced.

Authors reply – The authors have inserted ‘strength and conditioning’ here to introduce the acronym ‘S&C’.

MATERIALS AND METHODS

1. How have the participants been recruited?

Authors reply – The authors have included the following sentence: 

“Participants were recruited via social media advertisements which linked to an open weblink (Google, US), and in person during visits to boxing gyms in the North of England”.

2. Which has been the adherence, or the percentage of respondents compared to the potential and contacted participants?

Authors reply - The authors have now included the following information on the completion rate of the survey:

“A total of 101 completions translated to a completion rate (Number of boxers who answered the first question / number of boxers who completed the survey) of 88%.”

It was not possible to gauge the number of potential or contacted participants, as the survey was accessed voluntarily and anonymously.

3. Line 111: referring to the recruitments period, could Covid restriction have influenced the results?

Authors reply - The authors thank the reviewer for their interesting question. The amateur boxing season officially finishes in May, though it can typically finish in March or April for boxers who have not progressed to the finals of domestic tournaments. Covid restrictions came into law in the UK on 26th March 2020, whereby (thankfully) all survey data had already been collected, thus the pandemic did not influence data collection. 

4. Survey: in my opinion, the survey should be explained more in-depth; otherwise, it is difficult to understand all the variables explained in the results. For example, in the monitoring section, the authors should give more information about the option present in the survey and the possible responses (frequency or number of hours of the methods’ use, etc). Moreover, in the testing section, the authors should specify what they meant with LAB and FIELD. Finally, for the perception of the scientific support, it is better to specify that it is a Likert scale and the explanations of the points (see later for further details).

Authors reply – As per reviewer 1 suggestion, the authors have condensed the survey topic sections, into one paragraph. Specifically, authors include that weekly hours were collected as a measure of training volume and have also specified each training mode that were presented as options. This has been repeated for the monitoring and testing questions, with introduction of both ‘LAB’ and ‘FIELD’. Additionally, the authors agree that it is worth explicitly stating that the ‘perception of scientific support’ question is in Likert scale format, this has been done.

Lastly, the authors have supplied supplementary information (S1 Appendix) which highlights each question content and format, for further reader clarity. We hope the considerable changes have adequately addressed the reviewer concerns on survey information. 

5. Statistical analysis: why did the authors decide to utilize the ANOVA analysis? There are only two groups, and the authors even not evaluate neither main effects nor interactions (in the case of a two-way ANOVA).

Authors reply – As per reviewer 1 comment 9, the authors would like to reiterate their thanks in highlighting this error. The Friedman’s ANOVA was initially trialled as a suitable statistical test, where authors quickly decided it was not appropriate. Unfortunately, one sentence remained in text mistakenly. This has been deleted.

RESULTS

1. Lines 164-167: from which questions did the authors obtained these data?

Authors reply – The data relates to the following question that was initially in the physical training section: “Participants were also asked if they had access to a strength and conditioning coach (S&C) or sport scientist…”. 

The question can now be found in the first sentence of the Survey topics section. For further clarity, the authors have included a supplementary file (S1 Appendix) which details the question content and format (Please refer to ‘Access to performance staff’ - Q1).

2. Are there any other data, such as age, experience years, age at the onset of the sports practice, that could explain or be useful for the data interpretation?

Authors reply – The authors agree that the boxer background and experience information collected at the start of the survey would be useful for readers to interpret the data. The authors have included the following sentence when describing the subjects:

“One hundred and one amateur boxers completed the survey, including 59 SEB (Mean ± SD sex 50 males 9 females; age 20 ± 3 yrs; mass 71.5 ± 9.7 kg; training experience 7 ± 2 yrs) and 42 SDB (Mean ± SD sex 35 male 7 female; age 21 ± 4 yrs; mass 69.4 ± 9.9 kg; training experience 6 ± 2 yrs). “

3. The authors differentiated between bout and typical training week without specifying this in the statistical analysis or in the survey explanation (if the questions were double for bout and training week).

Authors reply - The following introduction has been added in the survey topic section “(typical training week or bout week)”. Likewise, the authors had initially included both typical training week and bout week data in the results section. For further clarity, the supplementary file (S1 Appendix) details each question, whereby it is clear to see the differentiation between typical training week and bout week.

4. Lines 178-179: the authors should not repeat data already present in the table. This also happens in other sections of the results.

Authors reply – The authors have removed results from the text where they are also present in the table.

5. Table 1: the authors could merge the columns of number and percentage in a single column (n (%)).

Authors reply – The authors agree with the reviewer’s suggestion, the columns have now been merged.

6. Line 201: in this line and in other sentences, the authors spoke about association; however, Mann-Withney and ANOVA evaluated the differences between the two groups and not the association. On the contrary, they should better specify the run analysis to understand why the wrote association.

Author reply – As per reviewer 1 comment 9, the inclusion of the Friedmans ANOVA sentence was an error. As the results section shows, no ANOVA data is, in fact, reported. We thank the reviewer for highlighting this error, and this has been deleted. The authors explored the associations between the dependent variables and boxer competitor level, whilst also providing descriptive statistics (Mean ± SD, %) to highlight differences between groups. 

Throughout the results section, the authors have ensured the reporting of the association between competitor level and the dependent variable. Where %’s have been reported for descriptive differences, the authors have now pre-empted this with “with descriptive data indicating…”. The authors now hope it is clear that the analysis in most questions is for an association, accompanied by descriptive data for differences. Select questions that were analysed via Mann Whitney, are for differences. This has been clearly stated for clarity.

7. Line 209-210: this question is not present in the survey explanation.

Author reply – These results were mistakenly left in the manuscript, in contrast, it was deleted from the survey explanation. Thank you for highlighting this error. The authors had agreed that this analysis did not have merit in this current work.

8. Lines 227-229: these data are not visible in Figure 1.

Author reply – The authors are not sure on this comment. Perhaps the reviewer meant figure 2? Figure 1 shows that 60% of SDB stated ‘none’ in reference to the question of monitoring intensity. Likewise, 32% of SEB stated none, therefore we feel the data is correct. However, we have made changes to the sentence structure:

“There was a significant association between competitor level and the monitoring of training intensity (χ2(1) = 7.449, P = 0.006, φ = .272), with descriptive data indicating that SDB were less likely to use a method to monitor intensity (40%) compared to SEB (68%) (fig 2).”

The authors have regularly presented descriptive percentages for ALL competitor level, for example “The most frequently used method for all boxers was HR monitoring (42%)”, as we feel this may be an interesting finding for the reader. We are; however, more than happy to delete this if the reviewer does not find it appropriate. 

9. Monitoring section: Data about RPE and none are not explained; however, many comments are present about this topic in the discussion section.

Authors reply – The authors are slightly confused about this comment, but we have tried our best to provide an appropriate action.

 If the reviewer means that RPE and ‘none’ should be explained in the survey topic section, this change has been made. 

If the reviewer is referring to the RPE and ‘none’ being explained in the results section, this has also been included. Data about ‘none’ refers to the following sentence:

“..with descriptive data indicating a reported lower application in SDB (40%) when compared to SEB (68%) (fig 2). 

Where 60% of SDB choose ‘none’, this translates to 40% of SDB using a form of monitoring method. Where 32% of SEB chose ‘none’, this translates to 68% of SEB using a form of monitoring method. 

The authors hope this adequately answers the above comment.

10. Testing: in figure 3, other significances are not commented on in the text.

Author reply – This has been added.

11. Lines 272-279: this question is not present in the survey explanation.

Authors reply – This has now been included and introduced in the survey topic section, prior to the results. 

DISCUSSION

1. Lines 293, 294, 319, 341, and 376-382 report data not present or different from the result section.

Author reply – The authors would like to emphasise that the data reported in lines 293 and 294 are indeed present in the results section, situated in the opening paragraph of the physical training section. The authors have made slight changes to ensure terms are presented similarly throughout. 

• Line 319 - as per reviewer 2 comment 7, this has been removed.

• Line 341 – Descriptive data from heart rate and RPE are now included in the results section, and therefore can also be referred to in the discussion. 

• Lines 376 – 382 – The authors would like to suggest that this data is already present in the results, in Fig 3. The following sentence has also been included in the results section, as the reporting of each individual percentage for both groups would impact on the word count of the manuscript: 

“Descriptive data demonstrated a consistently higher reported application in physical fitness testing in SEB when compared to their SDB counterparts (Fig 3).”

2. Line 361: the authors wrote influence, but they evaluated the difference.

Authors reply- As per previous comments, the authors have made it clearer throughout the manuscript that the primary analysis is association. Testing for differences was done in select questions, whilst descriptive data (Mean SD or %) was also provided in most cases for comparisons. 

In this specific case, and throughout the manuscript, the authors have aimed to be clearer on the terms used in relation to the statistical test used. For example, the above now reads:

“This work was also concerned with the association between competitor level and physical fitness testing in amateur boxing”.

 Likewise, the title of the manuscript has been changed, and now reads “The association between competitor level and the physical preparation practices of amateur boxers.”

3. Lines 366-368: the authors should specify that the data are not significant.

Authors reply - This change has been made.

4. Strengths and limitations are missing.

Authors reply – The authors have included a strengths and limitation section, highlighting the novelty of the work, but at the same time, addressing some of the drawbacks of the study that are common in survey-based research. The authors hope this is both informative and interesting for the reader.

---

## [Decision Letter · Decision Letter 1]

7 Sep 2021

**Comments to the Author**

1. If the authors have adequately addressed your comments raised in a previous round of review and you feel that this manuscript is now acceptable for publication, you may indicate that here to bypass the “Comments to the Author” section, enter your conflict of interest statement in the “Confidential to Editor” section, and submit your "Accept" recommendation.

Reviewer #1: All comments have been addressed

Reviewer #2: All comments have been addressed

2. Is the manuscript technically sound, and do the data support the conclusions?

Reviewer #1: Yes

Reviewer #2: Yes

3. Has the statistical analysis been performed appropriately and rigorously? 

Reviewer #1: Yes

Reviewer #2: Yes

4. Have the authors made all data underlying the findings in their manuscript fully available?

Reviewer #1: Yes

Reviewer #2: Yes

5. Is the manuscript presented in an intelligible fashion and written in standard English?

Reviewer #1: Yes

Reviewer #2: Yes

6. Review Comments to the Author

Reviewer #1: The authors have satisfied all doubts / criticisms, justifying the decision taken and modifying the requested parts. All this led to an improvement of the article compared to the previous version

Reviewer #2: Dear authors,

the quality and the understandability of the manuscript significantly improved after the revision.

I still have one doubt/curiosity.

Since the sample in composed by either males and females and the effort perceptions or structure or intensity of the training could be different in the two sexes, did you tried or would you try to adjust the analysis by sex to highlight if the sex variable is significant in your study? I strongly suggest to perform this analysis.

On the contrary, I suggest to add this point in the limitation section and consider sex differences in further studies.

7. PLOS authors have the option to publish the peer review history of their article (what does this mean?). If published, this will include your full peer review and any attached files.

Reviewer #1: **Yes: **Antonino Mulè

Reviewer #2: **Yes: **Lucia Castelli

---

## [Author Response · Author response to Decision Letter 1]

7 Sep 2021

Dear Emiliano Ce,

Thank you to you and your editorial team for the opportunity to submit a revised version of the manuscript. We are delighted that we have adequately responded to all reviewer comments. We would like to send our appreciation once again to the two reviewers for their time and expertise. In relation to the single suggestion provided by reviewer 2 that required minor revision, this has been addressed below. 

Once again, thank you.

Reviewer 2

1. Since the sample is composed by either males and females and the effort perceptions or structure of intensity of the training could be different in the two sexes, did you tried or would you try to adjust the analysis by sex to highlight if the sex variable is significant in your study? i strongly suggest to perform this analysis. on the contrary, I suggest to add this point in the limitation section and consider sex differences in further studies. 

Authors reply – The authors thank reviewer 2 for this suggestion. We had initially considered analysing the differences in practices by sex; however, due to the small sample of female participants, this analysis was not possible. The authors definitely agree that this is an interesting area and one worth investigating, particularly as female boxing continues to grow in popularity. The authors have now included this as a limitation, with a recommendation for further research to aim for a more proportionate sample size of males and females, in order to perform this analysis. Please see below.

“Future research may wish to aim for a more proportionate sample of males and females, in order to assess potential differences in the physical preparation practices between sexes.”

---

## [Decision Letter · Decision Letter 2]

14 Sep 2021

The Association between Competitor Level and the Physical Preparation Practices of Amateur Boxers.

PONE-D-21-10978R2

Dear Dr. Finlay,

We’re pleased to inform you that your manuscript has been judged scientifically suitable for publication and will be formally accepted for publication once it meets all outstanding technical requirements.

Kind regards,

Emiliano Cè

Academic Editor

PLOS ONE

Additional Editor Comments (optional):

Reviewers' comments:

Reviewer's Responses to Questions

**Comments to the Author**

1. If the authors have adequately addressed your comments raised in a previous round of review and you feel that this manuscript is now acceptable for publication, you may indicate that here to bypass the “Comments to the Author” section, enter your conflict of interest statement in the “Confidential to Editor” section, and submit your "Accept" recommendation.

Reviewer #1: (No Response)

Reviewer #2: All comments have been addressed

2. Is the manuscript technically sound, and do the data support the conclusions?

Reviewer #1: Yes

Reviewer #2: Yes

3. Has the statistical analysis been performed appropriately and rigorously? 

Reviewer #1: Yes

Reviewer #2: Yes

4. Have the authors made all data underlying the findings in their manuscript fully available?

Reviewer #1: Yes

Reviewer #2: No

5. Is the manuscript presented in an intelligible fashion and written in standard English?

Reviewer #1: Yes

Reviewer #2: Yes

6. Review Comments to the Author

Reviewer #1: (No Response)

Reviewer #2: (No Response)

7. PLOS authors have the option to publish the peer review history of their article (what does this mean?). If published, this will include your full peer review and any attached files.

Reviewer #1: **Yes: **Antonino Mulè

Reviewer #2: **Yes: **Lucia Castelli

---

## [Editor Report · Acceptance letter]

17 Sep 2021

PONE-D-21-10978R2 

The association between competitor level and the physical preparation practices of amateur boxers. 

Dear Dr. Finlay:

I'm pleased to inform you that your manuscript has been deemed suitable for publication in PLOS ONE. Congratulations! Your manuscript is now with our production department. 

Kind regards, 

on behalf of

Professor Emiliano Cè 

Academic Editor

PLOS ONE